# Lion: Adversarial Distillation of Proprietary Large Language Models

**Yuxin Jiang[1,2]    Chunkit Chan[2*]    Mingyang Chen[1,2*]    Wei Wang[1,2]**

[1]The Hong Kong University of Science and Technology (Guangzhou), Guangzhou, China
[2]The Hong Kong University of Science and Technology, Hong Kong SAR, China
{yjiangcm, ckchancc, mchenbt}@connect.ust.hk, weiwcs@ust.hk

## Abstract

The practice of transferring knowledge from a sophisticated, proprietary large language model (LLM) to a compact, open-source LLM has garnered considerable attention. Previous works have focused on a unidirectional knowledge distillation way by aligning the responses of the student model with those of the teacher models to a set of instructions. Nevertheless, they overlooked the possibility of incorporating any "feedback"—identifying challenging instructions where the student model's performance falls short—to boost the student model's proficiency iteratively. To this end, we propose a novel adversarial distillation framework for a more efficient knowledge transfer. Leveraging the versatile role adaptability of LLMs, we prompt the teacher model to identify "hard" instructions and generate new "hard" instructions for the student model, creating a three-stage adversarial loop of imitation, discrimination, and generation. By applying this adversarial framework, we successfully transfer knowledge from ChatGPT to a student model (named **Lion**), using a mere 70k training data. Our results show that Lion-13B not only achieves comparable open-ended generation capabilities to ChatGPT but surpasses conventional state-of-the-art (SOTA) instruction-tuned models like Vicuna-13B by 55.4% in challenging zero-shot reasoning benchmarks such as BIG-Bench Hard (BBH) and 16.7% on AGIEval.[1]

## 1 Introduction

Large language models (LLMs) capable of following natural language instructions have exhibited tremendous success in generalizing zero-shot to new tasks (Mishra et al., 2022; Wei et al., 2022a). Due to various concerns, the most advanced LLMs, such as ChatGPT (OpenAI, 2022) and GPT-4 (OpenAI, 2023) that boasting billions of parameters, are

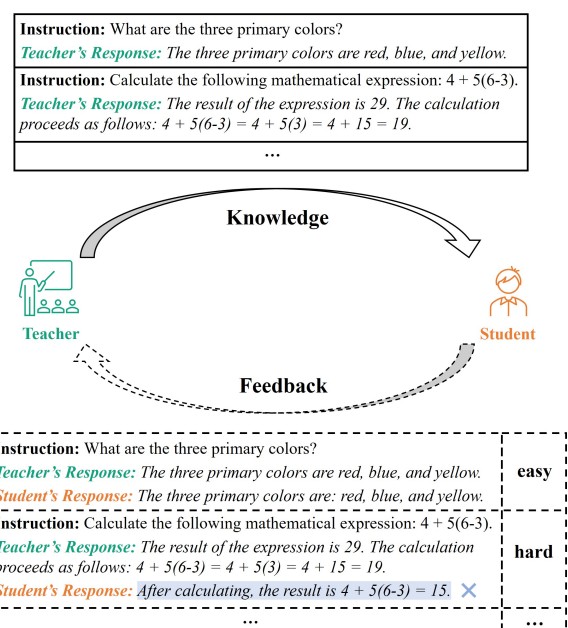

Figure 1: An illustration of the distinction between our approach and earlier ones. Previous methods facilitate a one-way knowledge transfer from the teacher to the student (*solid arrow*). Our approach, however, incorporates an innovative step (*dashed arrow*) that completes a loop: it enables the feedback"—identifying the student model's weaknesses—to be relayed back to the teacher, in order to foster tailored learning.

typically proprietary, comprising both the model parameter and the training data. To foster increased transparency regarding their intricate operational mechanics, a surge in research efforts focusing on knowledge distillation from a proprietary "teacher" LLM to an open-source "student" LLM. This is typically accomplished by aligning the responses of the student model with those of the teacher model to a set of instructions, which can be manually or automatically generated (Wang et al., 2022; Taori et al., 2023; Chiang et al., 2023; Xu et al., 2023).

However, previous works employ a unidirectional approach to knowledge transfer (solid arrow in Figure 1), where the teacher imparts knowledge to the student without considering any "feedback".

---

*The two authors have equal contributions.

[1]Code and model can be found at https://github.com/YJiangcm/Lion.

To better illustrate this using a tangible classroom scenario, the "feedback" refers to identifying the "hard" examples or problems where the student's performance falls short. This feedback guarantees that the teacher can provide bespoke training that centers on "hard" examples, thereby paving the way for more effective and tailored learning experiences for the student.

Inspired by adversarial knowledge distillation (AKD), which aims to iteratively improve the student model's performance by learning from generated hard samples (Fang et al., 2019; Micaelli and Storkey, 2019a; Heo et al., 2019), we propose an adversarial framework for distilling a proprietary LLM into a compact student model. Nevertheless, these AKD methodologies necessitate accessibility to the weights or gradients of the teacher model, which cannot be directly adapted to our setting. To circumvent this problem, we leverage the unparalleled role adaptability of LLMs, which can be effectively employed through a diverse range of prompts (Sanh et al., 2022). In particular, we prompt the proprietary teacher LLM to serve as a "referee" to discriminate hard instructions where there exists a significant performance discrepancy between the teacher's and student's responses, and serve as a "generator" to produce new instructions that emulate the data distributions corresponding to the discriminated hard instructions. Our framework, as depicted in Figure 2, consists of three stages in an iteration: 1) an imitation stage to align the student's response with the teacher's response; 2) a discrimination stage to identify hard instructions; 3) A generation stage to produce new hard instructions for escalating the challenges presented to the student model. In essence, our adversarial framework forms a *positive feedback loop* that efficiently bootstraps the student model's proficiency.

To verify the efficiency and efficacy of our method, we apply our AKD framework to transfer the knowledge of ChatGPT [2] onto an open-source foundation LLM, known as LLaMA (Touvron et al., 2023). We select Alpaca's training data (generated from only 175 manually selected seed instructions) as the initial training instructions and execute three iterations of AKD, resulting in a total of 70K data that our model is trained on. We've christened our model as **Lion**, drawing inspiration from the art of "distillation". By conducting extensive exper-

iments on open-ended generation and reasoning datasets, which include a total of 40 sub-tasks, our Lion-13B showcases superior performance surpassing instruction-tuned baseline models such as Vicuna (Chiang et al., 2023). Our main contributions are as follows:

- Our work is the first attempt to adopt the idea of adversarial knowledge distillation to large language models.

- Our proposed framework demonstrates impressive efficiency and efficacy. With instruction tuning performed on 70k data without any human annotation, our Lion-13B approximates ChatGPT's capabilities on open-ended generation dataset and largely outperforms the current SOTA model Vicuna-13B on reasoning tasks.

- The versatility of our framework allows for broad application: it is not exclusive to ChatGPT but can be conveniently adapted to suit a variety of other proprietary LLMs.

## 2 Related Work

### 2.1 Instruction-Following Language Models

With the impressive ability of instruction-following large language models such as ChatGPT (OpenAI, 2022) and GPT-4 (OpenAI, 2023), the techniques of instruction tuning (Wei et al., 2022b) have attracted a lot of attention (Wei et al., 2022c; Bubeck et al., 2023; Bang et al., 2023; Chan et al., 2023a). The early research of instruction tuning aims to enhance the generalization ability of language models, allowing these models to perform new tasks by comprehending task descriptions without relying on a few examplars. By fine-tuning these instruction-following language models (e.g., T5 (Raffel et al., 2020), FLAN (Aribandi et al., 2022), T0 (Sanh et al., 2022), and ExT5 (Aribandi et al., 2022)) on multi-task datasets in the form of natural language phrased as instructions, these models have been shown to perform well on unseen tasks with the instructions.

However, these models are only fine-tuned on simple task-specific instructions, and it is challenging to comprehend the sophisticated and diverse intent of users in real-world scenarios. Therefore, InstructGPT (Wei et al., 2022b), ChatGPT (OpenAI, 2022), and GPT-4 (OpenAI, 2023) trained on the diverse forms and abundant task types of

---

[2] We access ChatGPT using the OpenAI API (*gpt-3.5-turbo model*).

human-crafted instructions annotated by a considerable number of annotators. Since these instructions were not open-sourced, recent works such as Alpaca (Taori et al., 2023), Vicuna (Chiang et al., 2023), and WizardLM (Xu et al., 2023) investigate how to generate high-quality instructions and fine-tune the open-source large language model LLaMA (Touvron et al., 2023) with them to approach the performance of ChatGPT.

## 2.2 Knowledge Distillation

Knowledge Distillation (KD) (Hinton et al., 2015; Radosavovic et al., 2018; Chen et al., 2019) represents a crucial strategy within the sphere of model compression and acceleration, wherein a compact student model is instructed to emulate the performance traits of a more cumbersome teacher model. In practical contexts, the availability of training data is often constrained due to concerns regarding privacy, legality, security, or confidentiality. To address the absence of training data, data-free KD methods were proposed to align the student model to the teacher model, capitalizing on either related proxy data (Orekondy et al., 2019; Papernot et al., 2017) or synthetic data generated by learnable generators (e.g., Generative Adversarial Network (GAN)) (Addepalli et al., 2020; Fang et al., 2019; Micaelli and Storkey, 2019b) or teacher model inversions (Yin et al., 2020; Chawla et al., 2021; Fang et al., 2022). Nevertheless, these KD methodologies necessitate the accessibility to the weights or gradients of the teacher model. Consequently, an alternative line of research, commonly denoted as data-free model extraction (or stealing), endeavors to bridge this gap by employing zero-order estimation methodologies to approximate the authentic gradients of the teacher model to guide the update of the optimized generators (Kariyappa et al., 2021; Truong et al., 2021). However, adapting these methods to our distillation task presents two main hurdles. First, these techniques are primarily designed for image-based classification tasks, assuming access to a continuous softmax vector from the teacher model. Estimating zero-order gradients becomes problematic in our case, as responses are typically sequence-oriented. Second, developing an effective instruction generator capable of producing diverse, high-quality instructions that mirror the teacher model's training data distribution proves more challenging than in the image domain.

## 3 Methodology

Harnessing the learned knowledge of a sophisticated teacher model $\mathcal{T}(x; \theta^{\mathcal{T}})$ where the parameter $\theta^{\mathcal{T}}$ is inaccessible, our goal is to craft a more lightweight student model $\mathcal{S}(x; \theta^{\mathcal{S}})$. Ideally, a student model is optimal if the expectation of model discrepancy (which indicates the prediction differences between teacher $\mathcal{T}$ and student $\mathcal{S}$) on the uniform data distribution is minimized. Inspired by the success of adversarial knowledge distillation (AKD) (Fang et al., 2019; Micaelli and Storkey, 2019a; Heo et al., 2019), we turn to optimize an upper bound of the expectation —the expectation of the model discrepancy on "hard samples", where the teacher $\mathcal{T}$ and the student $\mathcal{S}$ have a relatively large performance gap. These "hard samples" are inclined to dominate the expectation of the model discrepancy. Thus, the overall expected model discrepancy can be effectively and efficiently reduced by optimizing the student model $\mathcal{S}$ on these "hard samples". The underlying rationale is rather straightforward and can be analogized to a real-world educational scenario: continuously concentrating on the "hard" knowledge that the student finds challenging to grasp is the most effective manner of enhancing a student's proficiency.

However, in the process of training the student model $\mathcal{S}$, hard samples will be mastered by the student and converted into easy samples. Hence we need a mechanism to continuously generate hard samples, which can be achieved by an adversarial framework.

The whole framework of our *Adversarial Knowledge Distillation* is depicted in Figure 2, which contains three stages in an iteration: 1) an imitation stage to align the student's response with the teacher's response; 2) a discrimination stage to identify hard samples; 3) A generation stage to produce new hard samples for escalating the challenges presented to the student model.

### 3.1 Initilization

As shown in Figure 2, four roles and two data pools are established in our framework, and we will comprehensively illustrate their functions later. We initialize our student model $\mathcal{S}$ using a foundation LLM such as LLaMA (Touvron et al., 2023). We initialize our teacher model $\mathcal{T}$, referee $\mathcal{R}$, and generator $\mathcal{G}$ by using the same proprietary LLM such as ChatGPT (OpenAI, 2022). The multiple roles that this proprietary LLM serves are accomplished

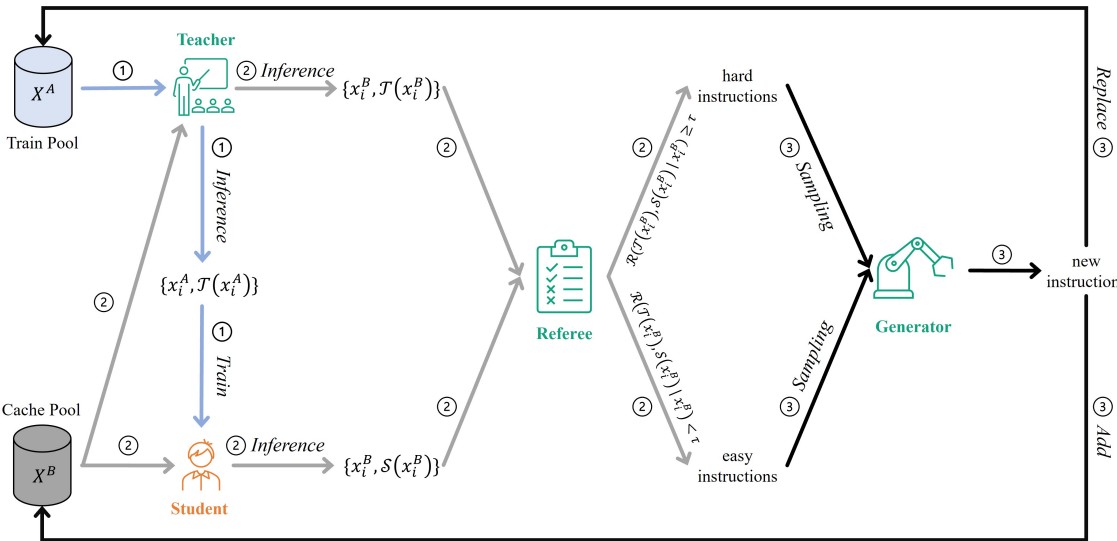

Figure 2: The overview of our adversarial distillation framework, where we craft a compact Student LLM $\mathcal{S}$ based on a superior proprietary LLM that serves three roles: the **Teacher** $\mathcal{T}$, the **Referee** $\mathcal{R}$, and the **Generator** $\mathcal{G}$. From left to right, there are three stages in an iteration: 1) Imitation; 2) Discrimination; 3) Generation.

through the use of varied prompt templates. We start the iteration from a given initial Train Pool $X^A = \{x_i^A\}_{i \in [1, N^A]}$, where $x_i^A$ is the $i$-th instruction in $X^A$, and $N^A$ is the number of samples in $X^A$. The Cache Pool $X^B$ is initialized as identical to $X^A$, consisting of instructions to evaluate the performance of $\mathcal{S}$ and $\mathcal{T}$.

## 3.2 Imitation Stage

To impart the knowledge of the teacher to the student, we construct the instruction-response data $\{x_i^A, \mathcal{T}(x_i^A)\}_{i \in [1, N^A]}$ by forward propagating instructions in the Train Pool $X^A$ through the teacher $\mathcal{T}$. The prompt template used for model inference is shown in Table 10. Like the imitation training of previous work (Taori et al., 2023; Chiang et al., 2023), we fine-tune our student model $\mathcal{S}$ to align the response of the teacher model, by optimizing the autoregressive language modeling objective.

## 3.3 Discrimination Stage

Figure 2 demonstrates that the discrimination stage starts from the Cache Pool, denoted as $X^B$. Even though this pool begins with the same initialization as the Train Pool, their uses diverge. The Train Pool is rejuvenated by replacing its existing instructions with freshly generated instructions, whereas the Cache Pool is enriched by incorporating these generated instructions. As a result, the growing storage capacity of the Cache Pool provides a more extensive space for evaluating the performance gap between teacher $\mathcal{T}$ and student $\mathcal{S}$. This allows for

more thorough detection of hard instructions.

In the discrimination stage, we ask the proprietary LLM to serve as a "referee", which quantifies the performance gap between $\mathcal{T}$ and $\mathcal{S}$. Specifically, we feed each instruction $x_i^B$ in the Cache Pool $X^B$ through both the teacher $\mathcal{T}$ and student $\mathcal{S}$ to generate the outputs $\mathcal{T}(x_i^B)$ and $\mathcal{S}(x_i^B)$, respectively. Then we ask the referee $\mathcal{R}$ to quantitatively measure the quality difference between teacher's response $\mathcal{T}(x_i^B)$ and student's response $\mathcal{S}(x_i^B)$, conditioned on $x_i^B$:

$$d_i = \mathcal{R}(\mathcal{T}(x_i^B), \mathcal{S}(x_i^B) \mid x_i^B) \qquad (1)$$

The above process is conducted by using the prompt template (as shown in Table 11) inspired by (Chiang et al., 2023), which requires the LLM to consider the helpfulness, relevance, accuracy, and level of detail of two responses and output two scores. To mitigate the positional bias (Wang et al., 2023) of the LLM referee, we conduct two runs by exchanging the positions of the teacher's response and the student's response and compute the final score as the average of the two runs. Then $d_i$ is calculated as the difference between the teacher's score and the student's score. By setting a threshold $\tau$ (1.0 used in our experiments), we discriminate hard instructions as those instructions with $d_i \geq \tau$, and the others are identified as easy ones. Figure 3b provides a clear and intuitive demonstration of which kinds of instructions are discriminated as hard in the first iteration. Compared with the instructions in the Cache Pool (Figure 3a), the dis-

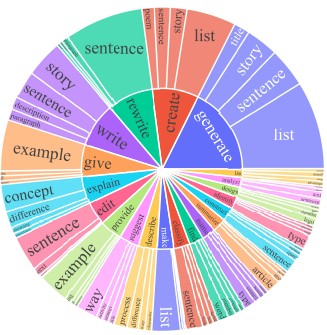

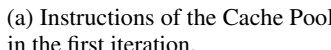
(a) Instructions of the Cache Pool in the first iteration.

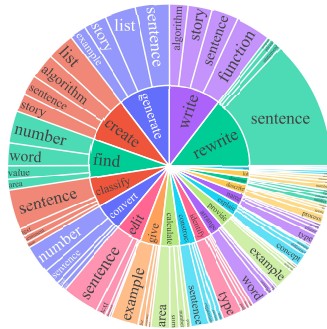

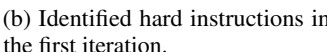
(b) Identified hard instructions in the first iteration.

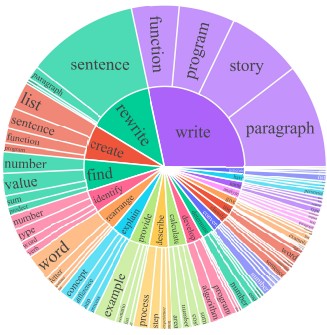
(c) Generated hard instructions in the first iteration.

Figure 3: The top 20 most common root verbs (inner circle) and their top 4 direct noun objects (outer circle) in the instructions.

tribution of the identified hard instructions is quite different, focusing more on complex tasks such as math, coding, etc.

### 3.4 Generation Stage

After carefully discerning the hard instructions, the generation stage aims to produce samples that mirror the data distributions corresponding to these challenging directives. This process is achieved by employing the proprietary LLM as a generator, denoted as $\mathcal{G}$, leveraging its exceptional prowess in content creation. Inspired by (Xu et al., 2023), we randomly sample an instruction from the hard instructions and prompt the generator $\mathcal{G}$ to generate a new instruction. The newly generated instruction is required to pertain to the same domain and match the task type of the sampled instruction. The template utilized for this prompt is exhibited in Table 12. As shown in Figure 3c, the distribution of the newly generated hard instructions appears to be comparable to that of the previously identified hard instructions. To mitigate the issue of catastrophic forgetting and to augment the diversity of the generated instructions, we also randomly sample an instruction from the easy instructions and prompt the generator $\mathcal{G}$ to generate a new instruction that belongs to the same domain as the sampled one, but exhibit a more long-tailed distribution. The template we use to prompt this process is displayed in Table 13.

In each iteration, we define $N$ as the total count of newly generated instructions and maintain a 1:1 ratio $r$ between the generated hard instructions and the generated easy instructions. To promote diversity, a new instruction will be deemed valid only if its ROUGE-L overlap with any existing instructions in the Cache Pool is below 0.7. Finally, as

aforementioned in Section 3.3, we proceed to rejuvenate the Train Pool, replacing its existing instructions with freshly generated ones. Concurrently, we enrich the Cache Pool by incorporating these newly generated instructions.

### 3.5 Min-Max Game Interpretation

Our adversarial knowledge distillation framework can be interpreted as a dynamic min-max game: in the imitation stage, we fine-tune our student to *minimize* the model discrepancy between itself and the teacher on hard samples; in the discrimination and generation stage, we craft new hard samples to *maximize* the model discrepancy, based on the learning progress of the student model. This dialectic framework propels the student model towards uncovering otherwise hidden knowledge, paving the way to complete understanding. As the training progresses through several iterations, the system should ideally achieve equilibrium. This is the point where the student model has mastered all the hard samples and the referee $\mathcal{R}$ can no longer distinguish between the student $\mathcal{S}$ and teacher $\mathcal{T}$ models. At this juncture, $\mathcal{S}$ becomes functionally indistinguishable from $\mathcal{T}$.

## 4 Experiments Setting

### 4.1 Datasets

In our experiments, we implemented a comprehensive LLM evaluation protocol that considers a diverse range of abilities, such as writing, coding, commonsense, math, and logical reasoning. The datasets we utilized can be classified into two main categories: open-ended generation and reasoning.

### 4.1.1 Open-ended Generation Datasets

**Vicuna-Instructions** (Chiang et al., 2023) is a set of 80 questions spanning 9 distinct task categories. This dataset has gained extensive usage in evaluating the capabilities of LLMs. Within our work, we examine LLMs' performance on this dataset in two different settings:

- **Setting1:** Following Vicuna (Chiang et al., 2023), we leverage GPT-4 to automatically assess the quality of responses (rated on a scale of 1 to 10) between a reference model (ChatGPT) and a candidate model. Subsequently, we calculate the candidate model's performance as the percentage of the total score it achieves compared to the reference model.

- **Setting2:** A recent work (Wang et al., 2023) pointed out that a systematic bias may exist in the above-mentioned GPT-4 automatic evaluation. To mitigate this, they propose two strategies, namely Multiple Evidence Calibration and Balanced Position Calibration, to obtain closer alignment with human judgments.

### 4.1.2 Reasoning Datasets

**AGIEval** (Zhong et al., 2023) is a well-known benchmark that quantifies the reasoning capability of foundation models in the context of human-centric standardized exams, including college entrance exams, math competitions, lawyer qualification tests, etc. We choose all English multiple-choice questions (8 tasks, 2,546 samples) among AGIEval for our experiments. The data statistics are shown in Table 6.

**BIG-Bench Hard (BBH)** (Suzgun et al., 2022) consists of a suite of challenging tasks from BIG-Bench (Srivastava et al., 2022), designed to assess the capabilities and limitations of large language models. These are the tasks on which prior language models underperform the average human rater. We choose all tasks that can be formatted into multiple-choice questions (23 tasks, 5,511 samples) among BBH for our experiments. The data statistics are shown in Table 7.

**Setting** We evaluate reasoning capabilities under a zero-shot setting without any exemplars and without Chain-of-Thought (CoT). For both AGIEval and BBH, we use the prompt format and parsing following (Zhong et al., 2023; Mukherjee et al.,

2023). Given the free-form response from the generative models, only the first capital character in the response is considered to compare with the gold answer (exact match). The result we report is accuracy (%).

### 4.2 Baselines

We select five superior LLMs as baselines, including LLaMA (Touvron et al., 2023), Alpaca (Taori et al., 2023), WizardLM (Xu et al., 2023), Vicuna (Chiang et al., 2023), and ChatGPT (OpenAI, 2022). It is worth noting that Vicuna has consistently ranked as the top open-source language model on multiple leaderboards, such as Chatbot Arena[3]. Therefore, we will conduct a comprehensive comparison with Vicuna. See detailed descriptions of these baselines in Appendix B.

### 4.3 Implementation Details

**Training Details** Our student model is initialized using the pre-trained LLaMA. The Train Pool and Cache Pool are initialized with the 52K automatically generated instructions from Alpaca (Taori et al., 2023). The total number of iterations is set to 3, with 6K newly generated instructions added at each iteration. This results in a total of 70K data that our model is trained on in order to make a fair comparison with current SOTA baselines, including WizardLM and Vicuna. The training hyperparameters are listed in Appendix C.

**Inference Details** To draw inferences from Lion and ChatGPT, we calibrated the temperature to 0.7 and set the maximum generation length at 1024. All other parameters adhere to their default settings. For LLaMA, Alpaca, WizardLM, and Vicuna, we configured their inference parameters in line with the specifications given in their respective original papers. When engaging with the gpt-3.5-turbo API for various roles, we employ an array of hyperparameters, the specifics of which can be located in Appendix C.

## 5 Experimental Results

### 5.1 Results for Open-ended Generation

Table 1 shows the performance comparison of various models against ChatGPT as the reference model, where GPT-4 is used as a referee/rater. Our Lion-7B and Lion-13B remarkably outperform their counterparts under two evaluation settings.

---

[3] https://chat.lmsys.org/?arena

| Model | Setting1 | Setting2 | Avg. |
|---|---|---|---|
| LLaMA-7B | 58.46 | 59.12 | 58.79 |
| Alpaca-7B | 69.29 | 67.20 | 68.25 |
| WizardLM-7B | 89.29 | 86.67 | 87.98 |
| Vicuna-7B | 87.79 | 89.96 | 88.88 |
| Lion-7B | **94.74** | **92.88** | **93.81** |
| LLaMA-13B | 69.23 | 68.21 | 68.72 |
| Alpaca-13B | 76.87 | 74.69 | 75.78 |
| Vicuna-13B | 92.25 | 92.97 | 92.61 |
| Lion-13B | **96.57** | **100.18** | **98.38** |

Table 1: Relative response quality (%) against ChatGPT (assessed by GPT-4) on Vicuna-Instructions.

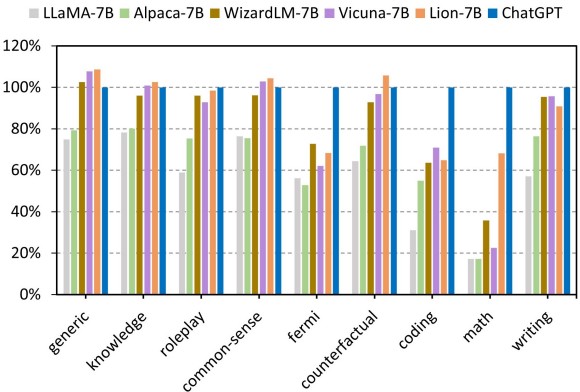

Figure 4: Relative response quality against ChatGPT on diverse task categories of Vicuna-Instructions.

Noticeably, Lion-13B shows an 8-point improvement over Vicuna-13B on aggregate, achieving 98.38% capabilities of ChatGPT.

To comprehensively compare with other baseline models on the capability to generate high-quality responses on various types of instruction, the relative response quality (Setting2) among different task categories is depicted in Figure 4. Our model impressively and slightly surpasses ChatGPT in the generic, knowledge, common-sense, and counterfactual task categories. Furthermore, for the two difficulty task categories described in the previous study (Chiang et al., 2023; Xu et al., 2023), our model significantly outperforms other baseline models with at least 32.32% relative score in the math task category while exceeding most of the baseline in the coding generation task category.

## 5.2 Results for Reasoning

**AGIEval Results** Table 2 presents the standard zero-shot performance comparison between Lion and baseline models on the AGIEval benchmark for multiple-choice English questions. Lion demonstrates significantly stronger performance compared to Vicuna, surpassing it in most task cate-

gories and achieving an average relative improvement of over 16%. However, Lion-13B still significantly lags behind ChatGPT, only retaining 72.5% of its reasoning capability.

**BIG-Bench Hard Results** Table 3 displays the zero-shot performance comparison between Lion and baseline models on BIG-Bench Hard with standard zero-shot prompting. Similar to AGIEval, Vicuna exhibits poor performance on sophisticated reasoning tasks within this benchmark, while Lion substantially surpasses Vicuna by around 50% on average. Particularly, Lion demonstrates significant performance enhancements of over 100% on tasks involving data understanding, semantic understanding (Disambiguation QA and Snarks), logical and geometric reasoning (Logical Deduction and Geometric Shapes), and position reasoning (Tracking Shuffled Objects). Despite achieving an average ability of nearly 74% compared to Chat-GPT on BBH, Lion-13B surpasses ChatGPT in several tasks, including Movie Recommendation, Snarks (identifying sarcastic sentences from two nearly-identical ones), and Tracking Shuffled Objects. This demonstrates the effectiveness of our method.

## 6 Analyses

### 6.1 Ablation Studies

**The threshold $\tau$ for distinguishing between hard and easy instructions** We systematically explored $\tau$ ranging from 0.0 to 2.0 and documented its influence on average performance across three datasets. Table 4 reveals an optimal range of $\tau$ between 1.0 and 1.5 for all datasets. Notably, elevating $\tau$ from 0.0 to 1.0 consistently enhances performance across all datasets, indicating effective differentiation between hard and easy instructions. However, a continuous increase from 1.0 to 2.0 gradually degrades performance due to decreased diversity in hard instructions. The ablation results demonstrate that our method is not quite sensitive to a large value of $\tau$.

**The ratio $r$ of generated hard and easy instructions** We change the ratio of generated hard instructions to generated easy instructions from 1:0 (all hard) to 0:1 (all easy) and investigate its impact on average performance across three datasets. It can be seen from Table 5 that higher ratios of hard to easy instructions generally lead to improved performance, with a balanced ratio of 1:1 yielding the

| Task | Human | | ChatGPT | Vicuna-7B | Lion-7B | Vicuna-13B | Lion-13B |
|---|---|---|---|---|---|---|---|
| | Avg | Top | | | | | |
| AQuA-RAT | 85.0 | 100.0 | 31.9 | **23.2** | 18.5 (-20.3%) | 20.1 | **26.0** (29.4%) |
| LogiQA | 86.0 | 95.0 | 35.0 | 21.4 | **31.8** (48.6%) | 29.8 | **31.3** (5.0%) |
| LSAT-AR | 56.0 | 91.0 | 24.4 | **22.2** | 17.4 (-21.6%) | 20.4 | **23.0** (12.7%) |
| LSAT-LR | 56.0 | 91.0 | 52.6 | 18.6 | **28.2** (51.6%) | 32.6 | **32.6** (0.0%) |
| LSAT-RC | 56.0 | 91.0 | 65.4 | 21.9 | **29.4** (34.2%) | 32.7 | **40.9** (25.1%) |
| SAT-Math | 66.0 | 94.0 | 42.7 | **21.4** | 20.9 (-2.3%) | 28.6 | **29.4** (2.8%) |
| SAT-English | 66.0 | 94.0 | 81.1 | 25.7 | **36.4** (41.6%) | 44.2 | **53.9** (21.9%) |
| SAT-English (w/o Psg.) | 66.0 | 94.0 | 44.2 | 26.2 | **27.7** (5.7%) | 26.2 | **36.2** (38.2%) |
| Average | 67.1 | 93.8 | 47.2 | 22.6 | **26.3** (16.4%) | 29.3 | **34.2** (16.7%) |

Table 2: Zero-shot performance comparison of ChatGPT, Vicuna, and Lion on AGIEval (multiple-choice English questions). We report the performance of Human, ChatGPT, and Vicuna from (Mukherjee et al., 2023). Performance improvements obtained by Lion over Vicuna are shown in parenthesis.

| Task | ChatGPT | Vicuna-7B | Lion-7B | Vicuna-13B | Lion-13B |
|---|---|---|---|---|---|
| Boolean Expressions | 82.8 | 39.2 | **55.2** (40.8%) | 40.8 | **65.6** (60.8%) |
| Causal Judgement | 57.2 | 39.7 | **50.3** (26.7%) | 42.2 | **43.9** (4.0%) |
| Date Understanding | 42.8 | 8.6 | **34.0** (295.3%) | 10.0 | **40.4** (304.0%) |
| Disambiguation QA | 57.2 | 15.2 | **35.6** (134.2%) | 18.4 | **44.8** (143.5%) |
| Formal Fallacies | 53.6 | 40.0 | **46.0** (15.0%) | 47.2 | **52.4** (11.0%) |
| Geometric Shapes | 25.6 | 3.6 | **8.8** (144.4%) | 3.6 | **8.8** (144.4%) |
| Hyperbaton | 69.2 | 42.8 | **51.6** (20.6%) | 44.0 | **56.8** (29.1%) |
| Logical Deduction (5 objects) | 38.8 | 4.8 | **19.6** (308.3%) | 4.8 | **20.8** (333.3%) |
| Logical Deduction (7 objects) | 39.6 | 1.2 | **14.4** (1100.0%) | 1.2 | **21.2** (1666.7%) |
| Logical Deduction (3 objects) | 60.4 | 19.6 | **40.4** (106.1%) | 16.8 | **38.0** (126.2%) |
| Movie Recommendation | 55.4 | 24.4 | **26.8** (9.8%) | 43.4 | **57.6** (32.7%) |
| Navigate | 55.6 | 43.6 | **49.2** (12.8%) | 46.4 | 45.2 (-2.6%) |
| Penguins in a Table | 45.9 | 17.5 | **24.7** (41.1%) | 15.1 | **26.7** (76.8%) |
| Reasoning about Colored Objects | 47.6 | 14.0 | **15.2** (8.6%) | 12.0 | **17.6** (46.7%) |
| Ruin Names | 56.0 | 12.2 | **14.4** (18.0%) | 15.7 | **29.2** (86.0%) |
| Salient Translation Error Detection | 40.8 | 2.0 | **12.0** (500.0%) | 2.0 | **12.4** (520.0%) |
| Snarks | 59.0 | 28.0 | **56.2** (100.7%) | 28.1 | **61.2** (117.8%) |
| Sports Understanding | 79.6 | 40.4 | **48.4** (19.8%) | 48.4 | **51.6** (6.6%) |
| Temporal Sequences | 35.6 | 21.2 | **24.4** (15.1%) | **16.0** | 10.4 (-35.0%) |
| Tracking Shuffled Objects (5 objects) | 18.4 | 6.4 | **14.4** (125.0%) | 9.2 | **24.8** (169.6%) |
| Tracking Shuffled Objects (7 objects) | 15.2 | 4.0 | **13.6** (240.0%) | 5.6 | **13.2** (135.7%) |
| Tracking Shuffled Objects (3 objects) | 31.6 | 26.8 | **34.0** (26.9%) | 23.2 | **34.4** (48.3%) |
| Web of Lies | 56.0 | **49.4** | 47.2 (-4.5%) | 41.2 | **54.8** (33.0%) |
| Average | 48.9 | 21.9 | **32.0** (45.9%) | 23.3 | **36.2** (55.4%) |

Table 3: Zero-shot performance comparison of ChatGPT, Vicuna, and Lion on BIGBench Hard (multiple-choice questions) without CoT. We report the performance of ChatGPT and Vicuna from (Mukherjee et al., 2023). Performance improvements obtained by Lion over Vicuna are shown in parenthesis.

highest average scores.

## 6.2 The Learning Dynamics of Lion

In Figure 5, we delve into the learning dynamics of Lion by visualizing its performance on AGIEval and BBH throughout the training iterations. The results clearly demonstrate that our adversarial knowledge distillation framework consistently enhances the performance of the student model as the iterations progress. Notably, the most significant improvement in capability occurs in the first iteration, suggesting the usefulness of the identification of challenging example patterns (refer Figure 3b).

## 6.3 Case Studies

To clearly compare the generated response quality between our model and other baselines, we provide nine case studies sampled from Vicuna-instruction, AGIEval, and BBH in Appendix E. Table 14 showcases the responses of various models to a math instruction. It can be seen that only Lion and ChatGPT provide the correct answer and follow the correct problem-solving steps. A counterfactual case is shown in Table 15, where ChatGPT provides a relevant answer that considers the potential impacts of Newton focusing on biology instead of physics, but it lacked details and depth. Lion, on

| Threshold $\tau$ | Vicuna-Instructions (Avg.) | AGIEval (Avg.) | BBH (Avg.) |
|---|---|---|---|
| 0.0 | 89.58 | 22.4 | 26.5 |
| 0.5 | 92.16 | 23.5 | 29.8 |
| 1.0 | 93.81 | **26.3** | **32.0** |
| 1.5 | **94.09** | 25.7 | 31.6 |
| 2.0 | 92.23 | 24.6 | 31.3 |

Table 4: Ablation study of the threshold $\tau$ for Lion-7B.

| Ratio $r$ | Vicuna-Instructions (Avg.) | AGIEval (Avg.) | BBH (Avg.) |
|---|---|---|---|
| 1:0 | 89.60 | 24.3 | 30.8 |
| 2:1 | 92.95 | 25.7 | **33.1** |
| 1:1 | **93.81** | **26.3** | 32.0 |
| 1:2 | 91.77 | 23.9 | 29.6 |
| 0:1 | 90.02 | 22.1 | 24.3 |

Table 5: Ablation study of the ratio $r$ for Lion-7B.

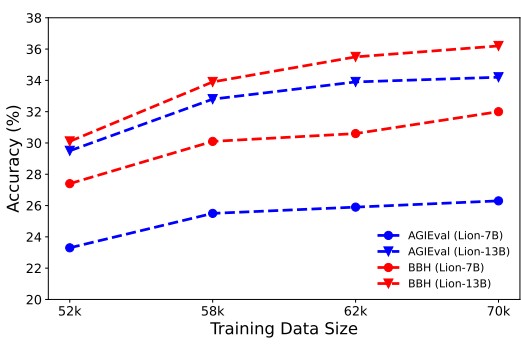

Figure 5: Performance of Lion-7B and Lion-13B on AGIEval and BBH through the training iterations.

the other hand, offered a more detailed and engaging response that explored different possibilities such as the development of biophysics or discovering new principles that could be applied to both fields. Lion's response also considered the potential implications of Newton's work on motion, force, gravity, and thermodynamics in biology, providing a more comprehensive answer.

## 7 Conclusion

This paper presents an innovative adversarial knowledge distillation framework for distilling a proprietary LLM into a compact, open-source student model. While previous methodologies have concentrated on unidirectional knowledge transfer, our approach seeks to integrate "feedback" into the learning process. Leveraging the versatile role adaptability of LLMs, we prompt the proprietary model to identify "hard" instructions and generate new "hard" instructions for the student model, creating a three-stage adversarial loop of imitation, discrimination, and generation. This approach al-

lows us to refine the student model's performance iteratively, efficiently bootstrapping its proficiency. We aspire that our model, named Lion, may serve as a baseline to reflect the performance of ChatGPT, especially the open-source instruction-following language model baseline for our community.

## Limitations and Discussions

**The Model Capability**  We have identified that Lion is subject to certain constraints: 1) A recent study (Gudibande et al., 2023) asserts that "model imitation is a false promise" since imitation models are adept at mimicking ChatGPT's style but fall short in improving LMs across more challenging tasks. While Lion still lags behind its teacher model ChatGPT in handling intricate reasoning tasks (as shown in our experiments), it demonstrates promising improvements compared to previous imitation models. Therefore, our adversarial knowledge distillation framework may provide a more effective way for knowledge transfer. 2) Since our training data doesn't encompass dialogues, Lion struggles to manage multi-turn conversations. 3) Due to computational resource constraints, Lion's maximum sequence length is limited to 1024. Consequently, it faces challenges when dealing with long documents. Despite these limitations, we envision Lion serving as an accessible springboard for future research endeavors aimed at addressing these limitations.

**The Training Process**  To train a single student model, we request the gpt-3.5-turbo API around 450k times, a number that is roughly 70% of the WizardLM's usage of 624k (Xu et al., 2023).

Nonetheless, this utilization incurs a considerable expense, nearing $900. In contrast to methods like Alpaca (Taori et al., 2023) and WizardLM (Xu et al., 2023), which only fine-tune the student model once, our adversarial knowledge distillation method employs iterative parametric updates to the student model. While this iterative approach inevitably leads to slower iteration speed, it offers additional benefits. Finally, different from traditional adversarial knowledge distillation where the weights of the generator are iteratively updated, we use a black-box and parameter-frozen LLM (ChatGPT in our paper) to serve the role. Therefore, the quality of the LLM is quite essential in the generation of new instructions.

**The Evaluation Metrics** Though automated evaluations leveraging GPT-4 have showcased promising prospects in appraising chatbot performance, the technique is yet to reach a level of maturity and accuracy, especially considering the propensity of large language models to generate non-existent or "hallucinated" information. Evaluating the efficacy of LLM across various tasks presents a considerable challenge since different tasks require quite different expertise (Wang et al., 2022). Therefore, the creation of a comprehensive, standardized evaluation system for chatbots is a prevailing research challenge that demands additional exploration and study.

### Ethics Statement

**Inherited Biases** It is important to consider that the behavior of our distilled student models may exhibit potential toxicity, biases, or privacy issues (Li et al., 2023a,b) inherited from the larger teacher LLM. We anticipate that the advancements made in reducing anti-social behaviors in LLMs can also be utilized to enhance student language models.

**License and Legality** Based on Stanford Alpaca's guidelines (Taori et al., 2023), we have determined that the weights of Lion will be exclusively licensed for research purposes in the future. Utilizing Lion's weights alongside LLaMA's original weights must adhere to Meta's LLaMA License Agreement. Users are responsible for acquiring and utilizing LLaMA in accordance with the license agreement.

**Safety** Unlike ChatGPT (OpenAI, 2022), Lion does not rely on human feedback to mitigate undesired behaviors. Instead, Lion learns to avoid such behaviors by imitating ChatGPT. However, it is important to acknowledge the potential risks associated with using Lion for malicious purposes, especially upon releasing its weights in the future. For future work, we aim to incorporate the technique of Reinforcement Learning from Human Feedback (RLHF) (Ouyang et al., 2022) to enhance access control. Additionally, Meta has implemented an access application process that can help regulate the distribution of LLaMA models and minimize the potential risks associated with their usage, providing an alternative option.

### Acknowledgements

W. Wang was also affiliated with Guangzhou Municipal Key Laboratory of Materials Informatics, The Hong Kong University of Science and Technology (Guangzhou), China. He was supported by HKUST(GZ) Grant G0101000028, GZU-HKUST Joint Research Collaboration Grant GZU22EG04, CCF-HuaweiDBC202302, and Guangzhou Municipal Science and Technology Project (No. 2023A03J0003).

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

## A  Data Statistics

Table 6 and Table 7 show the data statistics of AGIEval and BIG-Bench Hard, respectively.

| Task | # Examples | # Choices |
|---|---|---|
| AQuA-RAT | 254 | 5 |
| LogiQA | 651 | 4 |
| LSAT-AR | 230 | 5 |
| LSAT-LR | 510 | 5 |
| LSAT-RC | 269 | 5 |
| SAT-Math | 220 | 4 |
| SAT-English | 206 | 4 |
| SAT-English (w/o Psg.) | 206 | 4 |

Table 6: Statistics of AGIEval dataset.

| Task | # Examples | # Choices |
|---|---|---|
| Boolean Expressions | 250 | 2 |
| Causal Judgement | 187 | 2 |
| Date Understanding | 250 | 6 |
| Disambiguation QA | 250 | 4 |
| Formal Fallacies | 250 | 2 |
| Geometric Shapes | 250 | 11 |
| Hyperbaton | 250 | 2 |
| Logical Deduction (5 objects) | 250 | 5 |
| Logical Deduction (7 objects) | 250 | 7 |
| Logical Deduction (3 objects) | 250 | 3 |
| Movie Recommendation | 250 | 5 |
| Navigate | 250 | 2 |
| Penguins in a Table | 146 | 5 |
| Reasoning about Colored Objects | 250 | 18 |
| Ruin Names | 250 | 11 |
| Salient Translation Error Detection | 250 | 6 |
| Snarks | 178 | 2 |
| Sports Understanding | 250 | 2 |
| Temporal Sequences | 250 | 4 |
| Tracking Shuffled Objects (5 objects) | 250 | 5 |
| Tracking Shuffled Objects (7 objects) | 250 | 7 |
| Tracking Shuffled Objects (3 objects) | 250 | 3 |
| Web of Lies | 250 | 2 |

Table 7: Statistics of BIG-Bench Hard dataset.

## B  Baselines

- **LLaMA** (Touvron et al., 2023) is a collection of foundation language models ranging from 7B to 65B parameters. It is trained on trillions of tokens from publicly available datasets and is demonstrated to outperform larger-size LLMs such as GPT-3 (175B) across a multitude of benchmarks. We use the official code from LLaMA [4].

- **Alpaca** (Taori et al., 2023) is a project initiated by Stanford University with the objective of developing and disseminating an open-source model that adeptly follows instructions. It is based on LLaMA and fine-tuned on 52K instruction-following examples generated by

---

[4] https://github.com/facebookresearch/llama

querying OpenAI's text-davinci-003 model. On the self-instruct evaluation set, Alpaca mirrors text-davinci-003, but is notably more compact and cost-effective to reproduce. We use the official code from Alpaca [5].

- **WizardLM** (Xu et al., 2023) employs LLMs instead of humans to automatically mass-produce open-domain instructions of various difficulty levels, to improve the performance of LLMs. It uses an Evol-Instruct method to bootstrap the 52k instruction-following examples of Alapca into a larger set of 250k more intricate instructions. Out of this larger set, 70k examples were selected to fine-tune LLaMA. We use WizardLM-7B-V1.0 from the official code [6].

- **Vicuna** (Chiang et al., 2023), a superior open-source chatbot, excels in generating fluid and captivating responses to user queries. It is based on LLaMA and fine-tuned on 70K user-shared conversations collected from ShareGPT, a platform designed for sharing interactions with ChatGPT. Its impressive capabilities make it one of the leading open instruction-following models today. Vicuna achieves competitive performance against proprietary models such as ChatGPT and Bard (Google, 2023). We use Vicuna-7B-V1.1 and Vicuna-13B-V1.1 from FastChat [7].

- **ChatGPT** (OpenAI, 2022), a product of OpenAI, is an advanced AI chatbot renowned for its ability to interact with users in an authentically human and engaging manner. The chatbot is built on powerful LLMs such as GPT-3.5 and GPT-4, which are trained on a vast corpus of internet text data. ChatGPT undergoes fine-tuning via both supervised and reinforcement learning techniques, with the human trainers providing necessary feedback and direction.

## C Implementation Details

**Training Hyperparameters** The training process is conducted on 8 A100 GPUs. During each iteration of adversarial knowledge distillation, the hyperparameters for training are shown in Table 8.

---

[5] https://github.com/tatsu-lab/stanford_alpaca
[6] https://github.com/nlpxucan/WizardLM
[7] https://github.com/lm-sys/FastChat

| Hyperparameter | Lion-7B | Lion-13B |
|---|---|---|
| Batch size | 128 | 128 |
| Learning rate | 2e-5 | 2e-5 |
| Epoches | 3 | 3 |
| Max length | 1024 | 1024 |
| Optimizer | AdamW | AdamW |
| Scheduler | cosine | cosine |
| Weight decay | 0 | 0 |
| Warmup ratio | 0.03 | 0.03 |

Table 8: Training hyperparameters.

**Querying the gpt-3.5-turbo API** We use different sets of hyperparameters when querying the gpt-3.5-turbo API for different roles (Teacher, Referee, Generator). These hyperparameters are found to work well and we listed them in Table 9.

| Role | temperature | top_p | beam_size (n) | max_tokens |
|---|---|---|---|---|
| Teacher | 0.7 | 1.0 | 1 | 1024 |
| Referee | 0.2 | 1.0 | 1 | 512 |
| Generator | 1.0 | 1.0 | 1 | 512 |

Table 9: Hyperparameters for querying OpenAI gpt-3.5-turbo API under different roles.

## D Prompt Templates for Our Adversarial Distillation Framework

Fine-tuning an LLM (i.e. ChatGPT) is costly and intricate, human-tailored prompt templates are utilized to solve various tasks (Wei et al., 2022d; Chan et al., 2023b,c; Jiang et al., 2022; Jiayang et al., 2023; Chan and Chan, 2023). The prompt template of the **Teacher** for generating responses is shown in Table 10. The prompt template of the **Referee** for comparing the quality of two responses generated by two AI assistants is shown in Table 11. The prompt templates of the **Generator** for generating new hard instructions and new easy instructions are shown in Table 12 and Table 13, respectively.

## E Case Studies

Here we show 3 cases in Table 14, 15, and 16 to clearly compare the open-ended generation performance among various models including our Lion-13B, LLaMA-13B, Alpaca-13B, Vicuna-13B, and ChatGPT.

Besides, we show 6 cases in Table 17, 18, 19, 20, 21, and 22 to clearly compare the reasoning capability among various models including our Lion-13B, Vicuna-13B, and ChatGPT. We utilize ✓ and ✗ to denote whether the response is correct or incorrect, respectively.

| system content | You are a helpful assistant that generates a response to a given task instruction. |
|---|---|
| user content | ### Instruction:
{instruction}

### Response: |

Table 10: Prompt template of gpt-3.5-turbo for generating responses. Note that the original instruction in Alpaca is composed of an instruction prompt and an instance input. For example, the instruction prompt is "write an abstract about the following method", and the instance input is "knowledge distillation". For a better adaption to real-world scenarios, we concatenate the instruction prompt and the instruction prompt into one instruction using a line break.

| system content | You are a helpful and precise assistant for checking the quality of the answer. |
|---|---|
| user content | [Instruction]
{instruction}

[The Start of Assistant 1's Answer]
{answer_1}
[The End of Assistant 1's Answer]

[The Start of Assistant 2's Answer]
{answer_2}
[The End of Assistant 2's Answer]

[System]
We would like to request your feedback on the performance of two AI assistants in response to the user instruction and input displayed above.

Please rate the helpfulness, relevance, accuracy, and level of detail of their responses. Each assistant receives an overall score on a scale of 1 to 10, where a higher score indicates better overall performance.

Please first provide a comprehensive explanation of your evaluation, avoiding any potential bias and ensuring that the order in which the responses were presented does not affect your judgment. Then, output two lines indicating the scores for Assistant 1 and 2, respectively.

Output with the following format:
Evaluation evidence: <your evaluation explanation here>
Score of the Assistant 1: <score>
Score of the Assistant 2: <score> |

Table 11: Prompt template of gpt-3.5-turbo for comparing the quality of two responses generated by two AI assistants.

| system content | You are a helpful assistant. |
|---|---|
| user content | I want you to act as an Instruction Creator.
Your goal is to draw inspiration from the #Given Instruction# to create a brand new instruction.
This new instruction should belong to the same domain and the same task type as the #Given Instruction#.
The LENGTH and difficulty level of the #Created Instruction# should be similar to that of the #Given Instruction#.
The #Created Instruction# must be reasonable and must be understood and responded to by humans.
'#Given Instruction#', '#Created Instruction#', 'given instruction' and 'created instruction' are not allowed to appear in #Created Instruction#.

#Given Instruction#:
{instruction}

#Created Instruction#: |

Table 12: Prompt template of gpt-3.5-turbo for generating new hard instructions.

| system content | You are a helpful assistant. |
|---|---|
| user content | I want you to act as an Instruction Creator.
Your goal is to draw inspiration from the #Given Instruction# to create a brand new instruction.
This new instruction should belong to the same domain as the #Given Instruction# but be even more rare.
The LENGTH and difficulty level of the #Created Instruction# should be similar to that of the #Given Instruction#.
The #Created Instruction# must be reasonable and must be understood and responded to by humans.
'#Given Instruction#', '#Created Instruction#', 'given instruction' and 'created instruction' are not allowed to appear in #Created Instruction#.

#Given Instruction#:
{instruction}

#Created Instruction#: |

Table 13: Prompt template of gpt-3.5-turbo for generating new easy instructions.

| | |
|---|---|
| **Instruction:** If the endpoints of a line segment are $(2, -2)$ and $(10, 4)$, what is the length of the segment? | |
| **Lion** | To find the length of the line segment with endpoints $(2, -2)$ and $(10, 4)$, we can use the distance formula. The distance formula is derived from the Pythagorean theorem and is given as: $$d = \sqrt{((x2 - x1)^2 + (y2 - y1)^2)}$$ where $(x1, y1)$ and $(x2, y2)$ are the coordinates of the endpoints of the line segment. Substituting the given values, we get: $$d = \sqrt{((10 - 2)^2 + (4 - (-2))^2)}$$ Simplifying this expression, we get: $$d = \sqrt{((8)^2 + 6^2)}$$ $$d = \sqrt{(64 + 36)}$$ $$d = \sqrt{(100)}$$ $$d = 10$$ Therefore, the length of the line segment with endpoints $(2, -2)$ and $(10, 4)$ is 10 units. ✓ |
| **LLaMA** | The length of the line segment is 10 units. ✓ |
| **Alpaca** | The length of the line segment is 8 units. ✗ |
| **Vicuna** | The length of the line segment is the distance between the two points. To find the length, you can use the Pythagorean theorem, which states that the sum of the squares of the distances from a point to the two endpoints is equal to the square of the distance between the two endpoints. In this case, the distance from the point $(2, -2)$ to the point $(10, 4)$ is given by the formula: $$\sqrt{((x2 - x1)^2 + (y2 - y1)^2)}$$ $$= \sqrt{((10 - 2)^2 + (4 - (-2))^2)}$$ $$= \sqrt{(18^2 + 4^2)}$$ $$= \sqrt{(324 + 16)}$$ $$= \sqrt{(340)}$$ $$= 17.973$$ So the length of the line segment is 17.973. ✗ |
| **ChatGPT** | To find the length of a line segment with two endpoints $(x1, y1)$ and $(x2, y2)$, we use the distance formula: $$distance = \sqrt{((x2 - x1)^2 + (y2 - y1)^2)}$$ Plugging in the coordinates of the given endpoints, we get: $$distance = \sqrt{((10 - 2)^2 + (4 - (-2))^2)}$$ $$distance = \sqrt{(8^2 + 6^2)}$$ $$distance = \sqrt{(64 + 36)}$$ $$distance = \sqrt{(100)}$$ $$distance = 10$$ Therefore, the length of the line segment is 10 units. ✓ |

Table 14: A math case from Vicuna-Instructions.

| Instruction: | What if Isaac Newton had focused on biology instead of physics? |
| --- | --- |
| **Lion** | Isaac Newton is renowned for his contributions to physics, particularly his laws of motion and universal gravitation. However, what if Newton had instead chosen to focus on biology?

Newton's work in physics was heavily influenced by his fascination with the natural world and his desire to understand how it worked. Similarly, had he chosen to study biology, he may have made significant contributions to the field. Newton's mathematical abilities and his interest in the natural world would have made him a formidable biologist.

One area where Newton may have made a significant impact in biology is in the field of taxonomy. Newton was a keen observer of nature and had a particular interest in plants. He spent much of his time studying and cataloguing the various species of plants around him. He even developed a system for classifying plants based on their physical characteristics.

In addition to his work in taxonomy, Newton may have also made important contributions to the field of physiology. He was interested in the workings of the human body and may have studied the mechanics of muscles and the circulatory system.

Overall, while Isaac Newton is most well-known for his work in physics, his interests and abilities would have made him a formidable biologist as well. Who knows what discoveries he might have made had he chosen to focus on the natural world in a different way. |
| **LLaMA** | I'm not sure. If he did so, he wouldn't have discovered the law of gravity. |
| **Alpaca** | If Isaac Newton had focused on biology instead of physics, he might have made important contributions to the field of biology, such as developing a scientific theory of evolution or explaining the processes of cell growth and development. He might also have studied and discovered new principles of genetics and developed methods to study and treat diseases. |
| **Vicuna** | Isaac Newton is widely regarded as one of the most influential scientists in history for his work in physics, particularly for his development of the laws of motion and universal gravitation. However, it is worth noting that Newton also made significant contributions to other fields, including mathematics, optics, and natural philosophy.

If Newton had decided to focus on biology instead of physics, it is possible that he would have made significant contributions to the field. However, it is difficult to say exactly what those contributions would have been without knowing more about his specific interests and areas of focus.

Newton was a prolific scientist and wrote extensively on a wide range of topics, including natural philosophy, mathematics, and theology. He also conducted extensive research in optics, including the study of light and color, and made important contributions to the field of mathematics, including the development of calculus.

It is possible that if Newton had focused on biology, he would have made significant contributions to the understanding of living organisms and their behavior. However, it is also possible that his work in physics and mathematics would have had a greater impact on the world. |
| **ChatGPT** | If Isaac Newton had focused on biology instead of physics, the scientific world may be vastly different today. His advancements in mathematics, such as calculus, would have certainly been useful in the field of biology. Newton's curiosity, analytical mind, and dedication to scientific inquiry would have been equally effective in biology as they were in physics. He may have made significant contributions to the field of genetics or evolutionary theory.

On the other hand, his contributions to physics were so groundbreaking that the subject may have been hindered without his contributions. It's difficult to say exactly what would have happened, but it's likely that the course of scientific history would have been different. |

Table 15: A counterfactual case from Vicuna-Instructions.

| | **Instruction:** How can I develop my critical thinking skills? |
|---|---|
| **Lion** | Developing critical thinking skills is an important aspect of personal growth and professional development. Critical thinking enables us to analyze and evaluate information, ideas, and arguments objectively, make informed decisions, and solve complex problems. Here are some strategies that can help you develop your critical thinking skills:

1. Read widely and deeply: Reading widely exposes you to different perspectives and ideas, while reading deeply helps you to understand and analyze complex concepts. Choose books that are challenging and outside your comfort zone, and take notes as you read.

2. Ask questions: Asking questions is an essential part of critical thinking. Start by asking "why" and "how" questions to gain a deeper understanding of a topic. Then, ask "what if" questions to explore possible outcomes and consequences.

3. Practice analysis and evaluation: Critical thinking involves analyzing and evaluating information and arguments. Start by breaking down complex ideas into smaller parts, and then evaluate each part based on evidence, reasoning, and credibility.

4. Seek diverse perspectives: Surround yourself with people who have different backgrounds, experiences, and perspectives. Listen to their ideas and opinions, and challenge your own assumptions and biases.

5. Practice decision-making: Critical thinking is essential in decision-making. Practice making decisions by weighing the pros and cons, considering potential outcomes, and seeking advice from others.

6. Take courses and workshops: Enrolling in courses and workshops that focus on critical thinking, problem-solving, and decision-making can provide you with the tools and techniques to develop your critical thinking skills.

By implementing these strategies, you can develop your critical thinking skills and become better equipped to analyze, evaluate, and make informed decisions in your personal and professional life. |
| **LLaMA** | Critical thinking is a vital skill to have, but it can be difficult to develop. Here are some ways you can start developing your critical thinking skills:
1. Analyze and interpret: When you're presented with information, try to analyze and interpret it on your own terms. When you start to analyze and interpret information on your own, it will become easier to do so over time.
2. Assess your information: When you receive new information, try to assess whether it's valid or reliable. This is important for evaluating and forming your own opinions.
3. Use your intuition: Use your intuition to complement your critical thinking skills. When you pay attention to your gut feelings, they'll usually tell you when you need to double-check your information.
4. Learn from your mistakes: When you make a mistake, try not to beat yourself up about it. Instead, learn from your mistakes and use them to your advantage. |
| **Alpaca** | Developing critical thinking skills involves actively questioning assumptions, weighing evidence, and thinking analytically. It also involves being open-minded and considering various perspectives, as well as thinking logically and systematically. Practicing problem-solving and creative thinking, working through scenarios and dilemmas, and engaging in thoughtful conversations are all ways to improve critical thinking. Additionally, reading, reflecting, and writing can help increase your capacity for critical thinking. |
| **Vicuna** | Developing your critical thinking skills can help you to better analyze information, make more informed decisions, and solve complex problems. Here are some tips for improving your critical thinking skills:
1. Practice active listening: Pay attention to what others are saying and try to understand their perspective.
2. Ask questions: Be curious and ask questions to clarify information and better understand a situation.
3. Analyze information: Take time to carefully analyze information and consider different viewpoints.
4. Seek out diverse viewpoints: Expose yourself to a variety of viewpoints and ideas to challenge your own assumptions and biases.
5. Practice logic and reasoning: Use logic and reasoning to evaluate arguments and make decisions.
6. Learn from your mistakes: Be open to learning from your mistakes and using them as opportunities for growth.
7. Read and watch a variety of sources: Read and watch a variety of sources to expand your knowledge and expose yourself to different perspectives.
8. Engage in debates and discussions: Engage in debates and discussions with others to practice critical thinking and develop your skills.
9. Take breaks and reflect: Take breaks and reflect on your thinking process to improve your critical thinking skills.
10. Practice mindfulness: Mindfulness can help you to be more aware of your thoughts and biases, which can help you to think more critically. |
| **ChatGPT** | Sure, here are a few practical tips to develop critical thinking skills:
1. Ask questions: Don't simply accept information at face value. Ask questions that allow you to evaluate the reason, perspective and logic behind the information.

2. Analyze: Take the time to analyze and evaluate the information that you receive. This will help you to identify assumptions, biases, strengths and weaknesses of an argument.

3. Consider different perspectives: When analyzing information, try to consider different perspectives to gain a wider understanding of any topic.

4. Research and read extensively: Read a wide range of sources to get a better understanding of different viewpoints and opinions.

5. Practice logic and reasoning: Practice thinking through problems logically by breaking them down into smaller parts.

6. Establish a point of view: Consider developing an opinion or thesis statement, and then collect evidence to support that position.

7. Reflect on your own thinking: Consider how you arrived at a conclusion, and whether you have any biases or assumptions that may have influenced your thinking.

With time, and practicing these skills consistently, you can become a better critical thinker. |

Table 16: A generic case from Vicuna-Instructions.

| | **Instruction:** Q: If 120 is reduced to 96, what is the reduction percent? Answer Choices: (A)30% (B)40% (C)20% (D)10% (E)5% A: Among A through E, the answer is |
|---|---|
| **Lion** | To calculate the reduction percent, we need to divide the difference between the original number and the reduced number by the original number and then multiply by 100.

Here, the original number is 120 and the reduced number is 96. So, the difference between the two numbers is 120 - 96 = 24.

To calculate the percentage reduction, we divide the difference by the original number and then multiply by 100:

(24 / 120) x 100 = 20%

Therefore, the correct answer is option (C) 20%.

Option (A) 30%, option (B) 40%, option (D) 10%, and option (E) 5% are all incorrect. ✓ |
| **Vicuna** | A. 30% ✗ |
| **ChatGPT** | To calculate the reduction percentage, we need to find the difference between the original value and the reduced value, and then divide it by the original value.

The difference between 120 and 96 is 120 - 96 = 24.

Now, divide the difference by the original value and multiply by 100 to get the percentage reduction:

(24 / 120) * 100 = 20%

Therefore, the correct answer is (C) 20%. ✓ |

Table 17: An AQuA-RAT case from AGIEval.

| | **Instruction:** Q: not ( True ) and ( True ) is? Answer Choices: (A) True (B) False A: Among A through B, the answer is |
|---|---|
| **Lion** | The expression "not (True) and (True)" can be simplified as "(not True) and (True)" or "False and True". The "and" operator has a higher precedence than the "not" operator, so the first expression is evaluated as "False and True", which is false.

Therefore, the correct answer is option (B) False. ✓ |
| **Vicuna** | (B) False

### Explanation:

This statement is false.

If the given statement is "not (True)" and "True", it would be equivalent to "not (True) and True", which would be true.

(This is because negation of "True" is "False", and "True" and "False" evaluate to "True".)

Therefore, the correct answer is (A) True. ✗ |
| **ChatGPT** | (A) True. ✗ |

Table 18: A boolean expressions case from BIGBench Hard.

| | **Instruction:** Q: If you follow these instructions, do you return to the starting point? Take 8 steps. Turn around. Take 8 steps. Answer Choices: (A) Yes (B) No A: Among A through B, the answer is |
|---|---|
| **Lion** | According to the instructions given, if you follow them, you will return to the starting point. Therefore, the correct answer is Option A, "Yes". By taking 8 steps and turning around, you will end up back where you started. It is important to follow instructions carefully to achieve the desired outcome. In this case, following these instructions will ensure that you return to the starting point. ✓ |
| **Vicuna** | (A) Yes ✓ |
| **ChatGPT** | (B) No ✗ |

Table 19: A navigate case from BIGBench Hard.

| | Instruction: Q: Today is Christmas Eve of 1937. What is the date tomorrow in MM/DD/YYYY? Answer Choices: (A) 12/11/1937 (B) 12/25/1937 (C) 01/04/1938 (D) 12/04/1937 (E) 12/25/2006 (F) 07/25/1937 A: Among A through F, the answer is |
|---|---|
| **Lion** | The correct answer is (B) 12/25/1937.

To arrive at this answer, we need to consider the fact that Christmas Eve is the evening before Christmas Day. Therefore, the date of Christmas Day is one day after Christmas Eve. In this case, the date of Christmas Eve is 12/24/1937, and the date of Christmas Day is 12/25/1937. Therefore, the date tomorrow in MM/DD/YYYY format is 12/25/1937.

Option (A) is incorrect because it gives the date as 12/11/1937, which is incorrect.

Option (C) is incorrect because it gives the date as 01/04/1938, which is incorrect.

Option (D) is incorrect because it gives the date as 12/04/1937, which is incorrect.

Option (E) is incorrect because it gives the date as 12/25/2006, which is incorrect.

Option (F) is incorrect because it gives the date as 07/25/1937, which is incorrect. ✔ |
| **Vicuna** | (A) 12/11/1937
### Explanation:
The date tomorrow, on Christmas Eve of 1937, would be 12/11/1937. ✘ |
| **ChatGPT** | (C) 01/04/1938. ✘ |

Table 20: A data understanding case from BIGBench Hard.

| | Instruction: Q: Which statement is sarcastic? Answer Choices: (A) Hey just be happy then you won't be depressed anymore (B) Hey just be happy that you won't be depressed anymore A: Among A through B, the answer is |
|---|---|
| **Lion** | The correct answer is (B). The statement "Hey just be happy that you won't be depressed anymore" is sarcastic because it implies that being happy is an easy and straightforward solution to depression, which is a complex mental health issue that requires professional help and support. The use of the word "just" also adds a dismissive tone to the statement, suggesting that the speaker doesn't take depression seriously. It's important to avoid using sarcasm when talking about mental health issues, as it can be hurtful and invalidating to those who are struggling. Instead, it's important to offer compassion and support and to encourage individuals to seek professional help if needed. ✘ |
| **Vicuna** | (B) Hey just be happy that you won't be depressed anymore ✘ |
| **ChatGPT** | The sarcastic statement is (A) "Hey just be happy then you won't be depressed anymore." ✔ |

Table 21: A snarks case from BIGBench Hard.

| | Instruction: Q: In the following sentences, explain the antecedent of the pronoun (which thing the pronoun refers to), or state that it is ambiguous. Sentence: The scientist collaborated with the artist, and he shared a story. Answer Choices: (A) The scientist shared a story (B) The artist shared a story (C) Ambiguous A: Among A through C, the answer is |
|---|---|
| **Lion** | In the given sentence, the antecedent of the pronoun "he" is ambiguous. "He" could refer to either the scientist or the artist, as both of them shared a story. Therefore, option (C) is the correct answer. ✔ |
| **Vicuna** | (C) Ambiguous

### Explanation:

The pronoun "he" is ambiguous, as it could refer to either the scientist or the artist. Without more context, it is impossible to determine which person is being referred to. Therefore, the answer is "C - Ambiguous." ✔ |
| **ChatGPT** | (C) Ambiguous.

In the given sentence, the antecedent of the pronoun "he" is unclear. It could refer to either the scientist or the artist because there is no clear indication of which person shared the story. Therefore, the sentence is ambiguous. ✔ |

Table 22: A disambiguation QA case from BIGBench Hard.