# OpenReview forum: "Lion: Adversarial Distillation of Proprietary Large Language Models"
_EMNLP/2023/Conference — EMNLP 2023 Main_

### Official Review · Reviewer_XLBf · 2023-08-01

**Typos Grammar Style And Presentation Improvements:** The word "quantificationally" on line…
**Soundness:** 3

**Excitement:**

4: Strong: This paper deepens the understanding of some phenomenon or lowers the barriers to an existing research direction.

**Missing References:**

1. https://aclanthology.org/2021.emnlp-main.526.pdf
2. https://aclanthology.org/2021.acl-long.86/

**Paper Topic And Main Contributions:**

This paper presents a method for knowledge transfer from a large proprietary language model such as ChatGPT to a smaller open source model using model imitation. Motivated by Adversarial Knowledge Distillation, they discriminate between easy and hard instructions, sample from the hard ones and generate new instructions based on them and train on these in the next iterations. They also sample, generate and train from the easy ones to prevent catastrophic forgetting.

They demonstrate better performance compared to Vicuna, Alpaca and Llama model on different tasks and datasets.

**Questions For The Authors:**

1. This is not strictly knowledge distillation. Would it not be better to term it as adversarial data augmentation ?

2. The statement "our Lion-13B approximates ChatGPT’s open-ended generation capabilities" is a big claim and needs evidence. How do you support your statement ?

3. On Table 1, the Lion-13B on setting 2 shows a relative response quality of 100.18. Is this because GPT4 scores some Lion responses to be better than ChatGPT ?

**Reasons To Accept:**

1. Simple approach inspired from Adversarial KD which combines data augmentation and feedback from the LLM to improve on "harder" instructions.

2. Performance upgrade over similarly sized models.



**Reasons To Reject:**

1. Lack of an ablation study which can demonstrate sensitivity of hyperparameters as well as the value of augmentation vs feedback (discrimination)


**Reproducibility:**

4: Could mostly reproduce the results, but there may be some variation because of sample variance or minor variations in their interpretation of the protocol or method.

**Reviewer Confidence:**

4: Quite sure. I tried to check the important points carefully. It's unlikely, though conceivable, that I missed something that should affect my ratings.

---

> ### Author Rebuttal · Authors · 2023-08-28
>
> Thanks for your careful reading and detailed reviews, we hope the following comments could address your concerns.
>
> > Q1: Lack of an ablation study which can demonstrate sensitivity of hyperparameters as well as the value of augmentation vs feedback (discrimination).
>
> We have conducted a series of ablation studies based on Lion-7B, meticulously investigating the impact of various hyperparameters.
>
> - **The threshold $\tau$ for distinguishing between hard and easy instructions:** We systematically explored $\tau$ ranging from 0.0 to 2.0 and documented its influence on average performance across three datasets. Our findings reveal an optimal range of $\tau$ between 1.0 and 1.5 for all datasets. Notably, elevating $\tau$ from 0.0 to 1.0 consistently enhances performance across all datasets, indicating effective differentiation between hard and easy instructions. However, a continuous increase from 1.0 to 2.0 gradually degrades performance due to decreased diversity in hard instructions. The ablation results demonstrate that our method is not quite sensitive to a large value of $\tau$.
>
>     | $\tau$ | Vicuna-Instructions (Avg.)| AGIEval (Avg.) | BBH (Avg.) |
>     |:--------:|:-----:|:----------:|:---------:|
>     | 0.0 | 89.58 | 22.4 | 26.5 |
>     | 0.5 | 92.16 | 23.5 | 29.8 |
>     | 1.0 | 93.81 | **26.3** | **32.0** |
>     | 1.5 | **94.09** | 25.7 | 31.6 |
>     | 2.0 | 92.23 | 24.6 | 31.3 |
>
> - **The ratio of generated hard and easy instructions:** We change the ratio of generated hard instructions to generated easy instructions from 1:0 (all hard) to 0:1 (all easy) and investigate its impact on average performance across three datasets. It can be seen from the following table that higher ratios of hard to easy instructions generally lead to improved performance, with a balanced ratio of 1:1 yielding the highest average scores.
>
>     | Ratio of hard to easy | Vicuna-Instructions (Avg.)| AGIEval (Avg.) | BBH (Avg.) |
>     |:--------:|:-----:|:----------:|:---------:|
>     | 1:0 | 89.60 | 24.3 | 30.8 |
>     | 2:1 | 92.95 | 25.7 | **33.1** |
>     | 1:1 | **93.81** | **26.3** | 32.0 |
>     | 1:2 | 91.77 | 23.9 | 29.6 |
>     | 0:1 | 90.02 | 22.1 | 24.3 |
>
>
> > Q2: This is not strictly knowledge distillation. Would it not be better to term it as adversarial data augmentation?
>
> We understand your concern since our approach involves transferring knowledge from larger models to smaller ones through adversarially generated instructions, resembling data augmentation. Yet, it's important to consider that within the realm of proprietary Large Language Models (LLMs), the lack of accessible model parameters has led to data generation (augmentation) becoming an inherent aspect of knowledge distillation [1,2]. In this light, we propose that our method aligns with the concept of sequence-level knowledge distillation [3], where the content of distillation is in the form of an instruction task.
>
>
> > Q3: The statement "our Lion-13B approximates ChatGPT’s open-ended generation capabilities" is a big claim and needs evidence. How do you support your statement?
>
> The statement is supported by our experimental results. We evaluate Lion-13B on the widely adopted Vicuna-Instructions dataset (Lines 381-403), covering 9 varied task categories. Table 1 reveals Lion-13B's response quality at 96.57% and 100.18% relative to ChatGPT in distinct setups.
>
> However, we would like to highlight that evaluating the performance of large language models across various tasks is still an open problem (esp. in term of comprehensiveness and fairness). While our experimental findings contribute to this endeavor, we will further clarify this statement's scope in the final version.
>
>
> > Q4: On Table 1, the Lion-13B on setting 2 shows a relative response quality of 100.18. Is this because GPT4 scores some Lion responses to be better than ChatGPT?
>
> Yes, within setting 2, GPT-4 assigns superior scores to Lion-13B over ChatGPT in 52.5% of Vicuna-Instruction samples. Please refer to Appendix E for compelling case studies that further confirm Lion-13B's potential for generating superior responses in comparison to ChatGPT.
>
>
> > Typos and References
>
> Thank you for your detailed review. We will fix the typos and include all references you mentioned in our final version.
>
>
> References:
>
> [1] Self-Instruct: Aligning Language Models with Self-Generated Instructions. Wang et al. ACL 2023.
>
> [2] Large Language Models Can Self-Improve. Huang et al. arXiv 2022.
>
> [3] LaMini-LM: A Diverse Herd of Distilled Models from Large-Scale Instructions. Wu et al. arXiv 2023.

---

### Official Review · Reviewer_nhXy · 2023-08-05

**Soundness:** 4

**Excitement:**

3: Ambivalent: It has merits (e.g., it reports state-of-the-art results, the idea is nice), but there are key weaknesses (e.g., it describes incremental work), and it can significantly benefit from another round of revision. However, I won't object to accepting it if my co-reviewers champion it.

**Paper Topic And Main Contributions:**

This paper proposes a novel adversarial distillation framework for transferring a complex private large-scale language model (LLM) to a compact open-source LLM.By exploiting the versatile role adaptations of LLMs, the framework is able to motivate the teacher model to recognize "difficult" instructions and generate new "difficult" instructions for the student model, thus creating a three-stage adversarial loop. By utilizing the LLMs' versatile role adaptations, the framework is able to prompt the teacher model to identify "difficult" instructions and generate new "difficult" instructions for the student model, creating a three-stage adversarial cycle. By applying this adversarial framework, the authors successfully transferred knowledge from ChatGPT to a student model (named Lion) using only 70k of training data. Experimental results show that Lion-13B not only achieves a level comparable to ChatGPT in terms of open-ended generative power, but also outperforms traditional state-of-the-art models (SOTA) such as Vicuna-13B in challenging zero-sample reasoning benchmarks such as BIG-Bench Hard and AGIEval.

**Reasons To Accept:**

(1) Presents an innovative adversarial distillation framework that enables efficient knowledge transfer by modeling model differences between teacher models and student models.
(2) Extensive experiments on multiple open generative and reasoning benchmarks demonstrate the superior performance of the Lion model.
(3) It has a wide range of applications, not only for ChatGPT, but also provides a convenient knowledge transfer method for other proprietary LLMs.

**Reasons To Reject:**

(1) The selection of some parameters, such as the threshold τ for discriminating difficult and easy commands and the ratio of difficult and easy commands generated, is not discussed and analyzed in detail.
(2) The discussion of the model's capability, training process and evaluation metrics is rather brief and lacks in-depth analysis and discussion.

**Reproducibility:**

4: Could mostly reproduce the results, but there may be some variation because of sample variance or minor variations in their interpretation of the protocol or method.

**Reviewer Confidence:**

4: Quite sure. I tried to check the important points carefully. It's unlikely, though conceivable, that I missed something that should affect my ratings.

---

> ### Author Rebuttal · Authors · 2023-08-28
>
> Thank you for your review! We’re glad that you found our paper innovative, convincing, and adaptive.
>
> > Q1: The selection of some parameters, such as the threshold τ for discriminating difficult and easy commands and the ratio of difficult and easy commands generated, is not discussed and analyzed in detail.
>
> In response, we have conducted a series of ablation studies based on Lion-7B, meticulously investigating the impact of various hyperparameters.
>
> - **The threshold $\tau$ for distinguishing between hard and easy instructions:** We systematically explored $\tau$ ranging from 0.0 to 2.0 and documented its influence on average performance across three datasets. Our findings reveal an optimal range of $\tau$ between 1.0 and 1.5 for all datasets. Notably, elevating $\tau$ from 0.0 to 1.0 consistently enhances performance across all datasets, indicating effective differentiation between hard and easy instructions. However, a continuous increase from 1.0 to 2.0 gradually degrades performance due to decreased diversity in hard instructions. The ablation results demonstrate that our method is not quite sensitive to a large value of $\tau$.
>
>     | $\tau$ | Vicuna-Instructions (Avg.)| AGIEval (Avg.) | BBH (Avg.) |
>     |:--------:|:-----:|:----------:|:---------:|
>     | 0.0 | 89.58 | 22.4 | 26.5 |
>     | 0.5 | 92.16 | 23.5 | 29.8 |
>     | 1.0 | 93.81 | **26.3** | **32.0** |
>     | 1.5 | **94.09** | 25.7 | 31.6 |
>     | 2.0 | 92.23 | 24.6 | 31.3 |
>
> - **The ratio of generated hard and easy instructions:** We change the ratio of generated hard instructions to generated easy instructions from 1:0 (all hard) to 0:1 (all easy) and investigate its impact on average performance across three datasets. It can be seen from the following table that higher ratios of hard to easy instructions generally lead to improved performance, with a balanced ratio of 1:1 yielding the highest average scores.
>
>     | Ratio of hard to easy | Vicuna-Instructions (Avg.)| AGIEval (Avg.) | BBH (Avg.) |
>     |:--------:|:-----:|:----------:|:---------:|
>     | 1:0 | 89.60 | 24.3 | 30.8 |
>     | 2:1 | 92.95 | 25.7 | **33.1** |
>     | 1:1 | **93.81** | **26.3** | 32.0 |
>     | 1:2 | 91.77 | 23.9 | 29.6 |
>     | 0:1 | 90.02 | 22.1 | 24.3 |
>
>
> > Q2: The discussion of the model's capability, training process and evaluation metrics is rather brief and lacks in-depth analysis and discussion.
>
> We have provided a comprehensive discussion in the Limitation Section (Lines 565-629):
> - **Model Capability:** We have detailed Lion's limitations in handling reasoning, multi-turn conversations, and processing lengthy documents, along with corresponding explanatory insights.
> - **Training Process:** We demonstrate the cost-effectiveness of Lion's training through a comparative analysis with WizardLM. We acknowledge the trade-off of longer training times due to iterative parametric updates.
> - **Evaluation Metrics:** We emphasize concerns surrounding GPT-4's utilization in automated assessments and advocate for a robust, standardized evaluation framework for chatbots.
>
> Overall, we believe the ablation study and the discussion of limitation are minor points that do not dismiss the main claims supported in the paper. We are grateful for the opportunity to address these concerns and sincerely hope the reviewer will reevaluate the potential contribution of our work in light of the clarification provided.

---

### Official Review · Reviewer_THki · 2023-08-11

**Soundness:** 4

**Excitement:**

5: Transformative: This paper is likely to change its subfield or computational linguistics broadly. It should be considered for a best paper award. This paper changes the current understanding of some phenomenon, shows a widely held practice to be erroneous in someway, enables a promising direction of research for a (broad or narrow) topic, or creates an exciting new technique.

**Paper Topic And Main Contributions:**

This paper addresses the problem of the inaccessibility of proprietary large language models and presents a method (inspired by but distinct from previous work) to distill knowledge from a large language model like ChatGPT or LLaMA to a smaller model requiring far less data and no extra human annotation.  This paper provides contributions in the following areas: NLP engineering experiments, approach for data- and compute efficiency, and open source pre-trained models.

**Questions For The Authors:**

A) In terms of language modeling capabilities (expressability, creativity), how does Lion differ from ChatGPT or LLaMA?
B) Are the authors recommending that Lion can be used as a substitute for ChatGPT or LLaMA? If not, what are the authors recommending in terms of the use of Lion?
C) Are there potential risks of deploying Lion as a ChatGPT alternative (e.g., more instances of text degeneration phenomena like hallucination, length bias, repetition, copying, etc.) - asking as far as potential for these drawbacks (particularly so these could be mentioned in the paper); a thorough evaluation is not necessary.

**Reasons To Accept:**

This paper presents a new method (inspired by but distinct from previous work) to distill knowledge from ChatGPT or LLaMA to a smaller model requiring far less data and no extra human annotation.  The details of the method are presented in such a way that the paper is reproducible and the pitch is convincing.  The content and promise of the paper is exciting and likely of wide interest to the NLP community.   The experiments and evaluation is strong and covers a range of tasks and settings.

**Reasons To Reject:**

- The final two paragraphs of the introduction (Line 063-104) do not seem as carefully written as the rest of the paper. Particularly, there are quite a few instances in which the writing is vague; because of this, the text does not successfully serve the purpose it is meant to. I believe this could be remedied by revising the writing. Examples include: “forcing it to learn” (Line 065), “unparalleled role adaptability” (Line 074); generally, the overview of the method has vague phrases that make it difficult to understand.

- As the authors quote in the limitations section, “model imitation is a false promise” (Gudibande et al., 2023); however, the authors do not mention this in the main text of the paper.  Though the evaluation section is strong, beyond these settings, it is unclear from the paper itself what the potential drawbacks of this method are — for example, from a language modeling perspective, how does Lion differ from ChatGPT? Are the authors recommending that it can be used as a substitute, and if so, what are the potential drawbacks? A careful description of the pros and cons of the method and the difference in the end result from ChatGPT/LLaMA will clarify to future readers how this approach is best used and what contexts would be a inappropriate.

**Reproducibility:**

4: Could mostly reproduce the results, but there may be some variation because of sample variance or minor variations in their interpretation of the protocol or method.

**Reviewer Confidence:**

3: Pretty sure, but there's a chance I missed something. Although I have a good feel for this area in general, I did not carefully check the paper's details, e.g., the math, experimental design, or novelty.

**Typos Grammar Style And Presentation Improvements:**

- Figure 1 does not successfully clarify the difference between this approach and previous approaches.
- Figure 2: The caption or in-figure labels should be a little more verbose; for example, symbols/variables should be defined here as well.
- The method seems different from adversarial knowledge distillation; in fact, the use of “adversarial” throws me off a little bit in terms of my first impression vs. the content of the paper, so I would ask the authors to consider if they think it is valuable to emphasize the AKD method less and to make it clearer how their method either is different or how it fits with the AKD paradigm.

---

> ### Author Rebuttal · Authors · 2023-08-28
>
> Thank you for your review! We’re glad that you found our paper novel, convincing, and exciting.
>
> > Q1: In terms of language modeling capabilities (expressability, creativity), how does Lion differ from ChatGPT or LLaMA?
>
> Assessing the intricate aspects of expressability and creativity in language models remains challenging, often confined to limited text-based evaluations [1].
> Based on Margaret Boden's definition of creativity—“the ability to come up with ideas or artifacts that are *new*, *surprising* and *valuable*” [2], recent analysis has explored the creative potential of Large Language Models (LLMs), suggesting that while they achieve value and a modest level of novelty and surprise, their autoregressive nature hinders transformative creativity [3].
>
> Given that Lion, ChatGPT, and LLaMA share the autoregressive Transformer architecture, inherent limitations exist in their expressivity and creativity. Nonetheless, our empirical observations, exemplified by the counterfactual case in Table 13, indicate that Lion and ChatGPT may exhibit higher “expressivity” and “creativity” than LLaMA. This divergence could be attributed to further fine-tuning with instructional guidance.
>
>
> > Q2: Are the authors recommending that Lion can be used as a substitute for ChatGPT or LLaMA? If not, what are the authors recommending in terms of the use of Lion?
>
> Based on our extensive experiments (Table 1, 2, 3, and Figure 4), we advocate Lion as a viable alternative to ChatGPT in scenarios not requiring intricate reasoning, such as tasks involving general explanations (e.g., "Explain quantum computing basics") or practical operations (e.g., "Design a recipe to bake a chocolate cake"). Note that the Lion's lightweight nature and the ability to deploy locally presents several *practical* advantages for many applications scenarios, due to its resource efficiency, rapid inference speed, and far less concern in data security and privacy, compared with ChatGPT. However, we would like to highlight that evaluating the performance of large language models across various tasks is still an open problem (esp. in term of comprehensiveness and fairness); our experimental findings may serve as beneficial results towards this goal.
>
>
> > Q3: Are there potential risks of deploying Lion as a ChatGPT alternative?
>
> - **Security Issue:** As we mentioned (Lines 647-661), Lion does not utilize technics such as Reinforcement Learning from Human Feedback (RLHF) to reinforce the model to align with common values. Thus, it may exhibit unsafe output, and may be more vulnerable to certain attacks (e.g., prompt injection attack).
> - **Hallucination:** Lion may exhibit hallucination inherited from the larger teacher LLM (LLaMA in our paper). To investigate, our supplemental evaluation on TruthfulQA [4] reveals that while Lion-13B and LLaMA-13B exhibit comparable accuracy (46.5% vs. 47.0%), both fall behind ChatGPT (62.0%). Therefore, caution is advised in relying on Lion for critical decisions due to possible content fabrication.
> - **Length Bias:** Notably, Lion tends to produce lengthier responses than its base model LLaMA, as observed in Appendix E, Tables 12-14. This inclination might arise from Lion's training with hard instructions demanding increased response length.
>
> In the final version, we will include the above insights and additional case studies for a more comprehensive assessment.
>
>
> > Presentation Improvements
>
> Thanks for your valuable suggestions. We will remedy all the points mentioned according to your advice.
>
>
> References:
>
> [1] Putting GPT-3's Creativity to the (Alternative Uses) Test. Stevenson et al. ICCC 2022.
>
> [2] The Creative Mind: Myths and Mechanisms. M. A. Boden. Routledge 2003.
>
> [3] On the Creativity of Large Language Models. Franceschelli et al. arXiv 2023.
>
> [4] TruthfulQA: Measuring How Models Mimic Human Falsehoods. Lin et al. ACL 2022.

---

### Meta-Review · Area_Chair_TTui · 2023-09-19

**Recommendation:** 5

**Metareview:**

This paper addresses the problem of the inaccessibility of proprietary large language models and presents a method (inspired by but distinct from previous work) to distill knowledge from a large language model like ChatGPT or LLaMA to a smaller model requiring far less data and no extra human annotation.

Pros:

Presents an innovative adversarial distillation framework that enables efficient knowledge transfer by modeling model differences between teacher models and student models. It combines data augmentation and feedback from the LLM to improve on "harder" instructions.

Extensive experiments on multiple open generative and reasoning benchmarks demonstrate the superior performance of the Lion model.

Cons:

More ablation studies (some of them have been added by the authors during discussion/rebuttal)

Mor in-depth analysis/discussion on the model's training process and evaluation metrics

The original work  will also benefit from incorporating the additional discussion points/experiments conducted during rebuttal.

---

### Decision · Program_Chairs · 2023-10-07

**Decision:**

Accept-Main

**Comment:**

This paper addresses the problem of the inaccessibility of proprietary large language models and presents a method (inspired by but distinct from previous work) to distill knowledge from a large language model like ChatGPT or LLaMA to a smaller model requiring far less data and no extra human annotation.

Pros:

Presents an innovative adversarial distillation framework that enables efficient knowledge transfer by modeling model differences between teacher models and student models. It combines data augmentation and feedback from the LLM to improve on "harder" instructions.

Extensive experiments on multiple open generative and reasoning benchmarks demonstrate the superior performance of the Lion model.

Cons:

More ablation studies (some of them have been added by the authors during discussion/rebuttal)

Mor in-depth analysis/discussion on the model's training process and evaluation metrics

The original work  will also benefit from incorporating the additional discussion points/experiments conducted during rebuttal.